# Multi-Scale Feature Interactive Fusion Network for RGBT Tracking

**DOI:** 10.3390/s23073410

**Published:** 2023-03-24

**Authors:** Xianbing Xiao, Xingzhong Xiong, Fanqin Meng, Zhen Chen

**Affiliations:** 1School of Automation and Information Engineering, Sichuan University of Science and Engineering, Yibin 644000, China; 2Artificial Intelligence Key Laboratory of Sichuan Province, Sichuan University of Science and Engineering, Yibin 644000, China

**Keywords:** RGBT tracking, multi-scale feature, information interaction, transformer, attention mechanism

## Abstract

The fusion tracking of RGB and thermal infrared image (RGBT) is paid wide attention to due to their complementary advantages. Currently, most algorithms obtain modality weights through attention mechanisms to integrate multi-modalities information. They do not fully exploit the multi-scale information and ignore the rich contextual information among features, which limits the tracking performance to some extent. To solve this problem, this work proposes a new multi-scale feature interactive fusion network (MSIFNet) for RGBT tracking. Specifically, we use different convolution branches for multi-scale feature extraction and aggregate them through the feature selection module adaptively. At the same time, a Transformer interactive fusion module is proposed to build long-distance dependencies and enhance semantic representation further. Finally, a global feature fusion module is designed to adjust the global information adaptively. Numerous experiments on publicly available GTOT, RGBT234, and LasHeR datasets show that our algorithm outperforms the current mainstream tracking algorithms.

## 1. Introduction

Visual object tracking [1,2] is widely used in autonomous driving, medical diagnostics, traffic monitoring, and other fields. Visual tracking aims at estimating the position and scale of the object in subsequent frames, given its state in the first frame. Although many RGB-based tracking algorithms have achieved excellent results, the performance of these trackers still needs to be further improved in some challenging scenarios. For example, the poor imaging quality of RGB in severe weather conditions usually leads to tracking failures. Therefore, RGBT tracking, which can make use of the advantages of RGB and infrared, has received extensive attention. Specifically, the infrared image is formed by thermal radiation with the benefits of insensitivity to light and intense penetration [3]. RGB images have rich texture and color information. Therefore, RGBT tracking can better handle complex scenarios due to its complementary advantages. Complementary advantages are shown in Figure 1.

To obtain more accurate and robust target attribute information and compensate for the information uncertainty of a single modality in target tracking, it is necessary to fuse the data collected by multi-sensors. The existing fusion tracking methods can be divided into pixel-level, feature-level, and decision-level. Pixel-level fusion tracking means that heterogeneous images are fused first, then target tracking is carried out based on the rich images [4,5]. Although rich information is preserved, it will bring greater computational costs during tracking. In feature-level fusion tracking, the features of RGB and infrared images are first extracted and then fused according to the designed fusion rules to obtain the fused feature and finally use the fused feature to perform tracking [6,7,8]. Decision-level fusion tracking is first performed in individual modalities to obtain tracking results or response maps. The results or response maps are then fused to get the final tracking result [9,10]. The computational cost is relatively low compared to pixel-level and feature-level fusion tracking methods.

Although RGBT tracking has made significant progress, the mainstream methods can be roughly divided into two categories, namely the combined fusion tracking method and the discriminant fusion tracking method. The combined fusion tracking method aims to mine all valuable information of different modalities and combine them to achieve a richer representation. Li et al. [11] proposed a multi-adapter network to exploit modality-shared and modality-specific features, composed of a generic adapter, a modality adapter, and an instance adapter. Li et al. [12] proposed a challenge-aware network, summarizing existing datasets into several challenge attributes and training challenge branches one by one, which effectively enhanced the feature discrimination ability of poor modality. Although combined fusion is an effective fusion method, it tends to introduce noise and redundant information. However, the discriminant fusion tracking method aims to obtain discriminant information and achieve an effective fusion of information. To eliminate redundant information, Zhu et al. [13] proposed a fusion method based on the pruning strategy, using global average pooling and weighted random selection operations to score each channel and select high-quality channels for tracking. Zhang et al. [6] used the attention mechanism to calculate the reliability weights of multi-layer features to fuse multi-level deep features. Xu et al. [14] proposed a multi-modalities cross-layer bilinear pooling network, which uses channel and spatial attention mechanisms to predict the reliable weight of each position. However, these algorithms do not fully mine multi-scale information, limiting tracking performance when the target scale varies greatly. Moreover, they ignore the rich contextual information between features, and the interactive fusion of features can achieve more robust representations.

In response to the above problems, inspired by inception modules [15,16,17], we proposed a multi-scale feature interactive fusion network for RGBT tracking. Specifically, we add multiple branches to each layer for multi-scale feature extraction. In order to achieve effective fusion, we design a feature selection module that uses the channel-aware mechanism to calculate reliability weights for each branch and adaptively aggregate features from multiple branches. At the same time, the encoder and decoder of the Transformer are used to achieve self-enhancement and interaction enhancement. In addition, we designed a global feature fusion module to balance the contribution of different modalities in different scenarios, which adaptively adjusts global features in spatial and channel dimensions. Finally, three fully connected layers are utilized for instance learning and target state estimation.

To sum up, the main contributions of this work are as follows:We propose a new multi-scale feature interactive fusion network (MSIFNet) to implement robust RGBT tracking. The network can improve the recognition ability of targets of different sizes by fully exploiting multi-scale information, thus improving tracking accuracy and robustness;We design a feature selection module that adaptively selects multi-scale features for fusion by the channel-aware mechanism while effectively suppressing noise and redundant information brought by multiple branches;We propose a Transformer interactive fusion module to further enhance the aggregated feature and modality-specific features. It improves long-distance feature association and enhances semantic representation by exploring rich contextual information between features;We design a global feature fusion module, which adaptively adjusts the global information in spatial and channel dimensions, respectively, to integrate the global information more effectively.

## 2. Related Work

Visual tracking is a basic computer vision task, and many excellent algorithms have been proposed. This chapter will introduce relevant work from the following two aspects: (1) RGB tracker and (2) RGBT tracker.

(1)RGB trackers

Based on the correlation filter tracking algorithm [18,19], it has attracted ascendant attention because of its real-time speed, but handcrafted features limit the recognition ability. In response to this problem, siamFC [20] adopts the Siamese structure to introduce deep learning into tracking. With the application of Siamese networks in the tracking field, RGB tracking algorithms have developed rapidly. Additionally, a large number of algorithms based on Siamese networks have emerged, for example, siamRPN [21], SiamRPN++ [22] based on anchor, and siamFC++ [23], siamBAN [24], SiamCAR [25] based on the anchor-free mechanism. To optimize the tracking algorithm, Danelljan et al. [26] used off-line training IoU predictor and online training classifier for target state estimation, and they performed classification and regression tasks simultaneously in the tracking process to achieve robust tracking performance. To take advantage of the powerful feature expression ability of the Transformer, Wang et al. [27] used the Transformer to construct temporal information at different moments to model global dependencies better. Mayer et al. [28] aimed at the optimization-based tracking method that limits the expressive ability of the tracking network, used the Transformer to capture global relationships, and learned a more robust target prediction model.

(2)RGBT trackers

Traditional-based methods: Traditional RGBT tracking methods are mainly divided based on sparse representation and graph models. Since sparse representation can suppress noise and errors, early methods mainly focus on sparse representation. For example, Wu et al. [29] applied sparse representation to RGBT tracking for the first time, which integrated image patches of different modalities into a one-dimensional vector and carried out sparse representation in the target template. Li et al. [30] designed a fusion method based on Bayesian filtering, which considers intra-modality and inter-modality constraints for cross-modality sparse representation. Lan et al. [31] designed a modality-correlation-aware sparse representation model, adaptively selecting representative particles via low-rank and sparse regularization to handle appearance variations. The graph-based method can suppress the influence of noise background and has also received certain attention. To consider the synergism and heterogeneity between modalities, Li et al. [32] designed the cross-modality sorting graph model, introducing the cross-modality soft consistency to integrate multi-modalities information effectively. To eliminate background clutter, Shen et al. [33] proposed a cooperative low-rank graph model, which decomposes the input features into low-rank feature parts and sparse noise parts and uses the cooperative graph learning algorithm to renew dynamically. However, these works utilize handcrafted features for tracking, which limits the performance to handle various challenges.

Deep learning-based methods: Deep learning is known for its robust feature expressiveness, which can model the appearance of objects better than hand-crafted methods. Xu et al. [34] first applied deep learning to the field of RGBT tracking. Subsequently, deep RGBT trackers have dominated. DAPNet [13] performs fusion at different feature levels using the same aggregation network. DAFNet [35] also adopts a similar strategy to DAPNet for feature fusion. CMPP [36] utilizes the attention mechanism to model correlations between heterogeneous data. Considering the importance of time information in the video sequence, the time context information is correlated to achieve more effective information inheritance. CAT [12], ADRNet figure [37], and APFNet [38] all model robust appearance representations by training multiple challenge-aware branches independently. However, the implementation details are different, CAT [12] adaptively aggregates multiple challenge branches and then adds them to the backbone feature learning process. ADRNet designs an attribute-driven residual branch that models different challenge attributes and aggregates them through residual connections to obtain a powerful target representation. In APFNet, different challenge branches are aggregated by SKNet [39] adaptively and use Transformers to enhance features.

The above trackers perform well, but these networks do not explore multi-scale information, limiting tracking performance when the target scale varies greatly. To solve the above problems, we design a new multi-scale feature interactive fusion network to handle the RGBT tracking task.

## 3. Methods

This section will introduce the proposed multi-scale feature interactive fusion network. First, we outline the entire architecture, and then we present the structure of the individual module.

### 3.1. Network Architecture

As shown in Figure 2, our network uses a symmetrical parallel structure to mine the potential information of two modalities. Specifically, we used the VGG-M [40] as the backbone feature extraction network. To trade off speed and accuracy, we used only the first 3 layers, with convolution kernel sizes of 7 × 7, 5 × 5, and 3 × 3. Following the first and second layer convolution are the Relu function, local response normalization, and max-pooling. After the third layer convolution, only the Relu function is used.

Different convolution kernels have various receptive fields, and different receptive fields can better adapt to targets of different sizes. Based on this knowledge, we add 5 × 5 and 3 × 3 convolution branches in the first layer to extract multi-scale features and add 3 × 3 and 1 × 1 convolution branches in the second layer. Considering the third layer of the backbone network is a small convolution kernel, only a 1 × 1 convolution branch is added to the third layer. For feature maps of different sizes, we use max-pooling to fix feature maps to the exact resolution for better fusion. To better integrate multi-scale features, a feature selection module was designed to adaptively activate features from different branches for aggregation while eliminating noise and redundancy. Then, the aggregated feature and modality-specific features are fed into the Transformer interactive fusion module for interaction enhancement. After fusion, the enhanced feature is added to backbone features for subsequent feature extraction. Furthermore, we designed a gated network to block noise propagation. Finally, the global feature fusion module is used to further balance the contributions of different modalities in different scenarios, and three fully connected layers and a softmax layer are used to predict the position of the target.

In MANet [11], modality-shared and modality-specific features are helpful for modeling target information. To take full advantage of shared and specific features, our backbone network does not share parameters to extract modality-specific features, while other pairs of convolutions (e.g., 2 convolution branches of 5 × 5 in the first layer) share parameters to extract shared information.

### 3.2. Feature Selection Module

There is redundant information in the feature maps of multiple branches. The reliability of features should be considered before fusion. Inspired by SKNet [39], we designed the feature selection module (FSM) to get more feature clues by deploying a small number of parameters. The FSM can adaptively activate valuable features for fusion while effectively suppressing noise and redundant information. Specifically, the FSM first performs simple element-wise summation, then uses the global average pooling operation (GAP) to obtain the channel vector. Then, feed the channel vector into 2 convolutions of 1 × 1 and a softmax function to obtain the channel weights. Finally, the original features are weighted with channel weights. The FSM adaptively activates features from different branches, combining them to focus the network on beneficial information. The specific situation of feature selection is shown in Figure 3. The whole process is summarized as follows:(1)$$X^{a}=\left\{\begin{array}{cc} X_{5\times 5}^{i}⊕X_{5\times 5}^{v}⊕X_{3\times 3}^{i}⊕X_{3\times 3}^{v}, & l=1\\ X_{1\times 1}^{i}⊕X_{1\times 1}^{v}⊕X_{3\times 3}^{i}⊕X_{3\times 3}^{v}, & l=2\\ X_{1\times 1}^{i}⊕X_{1\times 1}^{v}, & l=3 \end{array}\right.$$
(2)$$w_{i}=\phi \langle \delta \langle f\left(GAP\left(X^{a}\right)\right)\rangle \rangle$$
(3)$$X^{sel}=\left\{\begin{array}{cc} w_{5\times 5}^{i}⊙X_{5\times 5}^{i}+w_{5\times 5}^{v}⊙X_{5\times 5}^{v}+w_{3\times 3}^{i}⊙X_{3\times 3}^{i}+w_{3\times 3}^{v}⊙X_{3\times 3}^{v}, & l=1\\ w_{3\times 3}^{i}⊙X_{3\times 3}^{i}+w_{3\times 3}^{v}⊙X_{3\times 3}^{v}+w_{1\times 1}^{i}⊙X_{1\times 1}^{i}+w_{1\times 1}^{v}⊙X_{1\times 1}^{v}, & l=2\\ w_{1\times 1}^{i}⊙X_{1\times 1}^{i}+w_{1\times 1}^{v}⊙X_{1\times 1}^{v}, & l=3 \end{array}\right.$$
where l represents the l-th layer. f denotes 2 convolution operations of 1 × 1 inlaid with the ReLU function, expressed as $f=f_{2}\left(\xi \left(f_{1}\right)\right)$. $f_{1}$ and $f_{2}$ represent 1 × 1 convolution operations. ξ and δ denote the ReLU function and softmax function, respectively. ϕ represents the reshape operation. $w_{i}$ represents the generated weight. ⊙ and ⊕ denote the element-wise product and element-wise summation.

### 3.3. Transformer Interactive Fusion Module

The superior performance of the Transformer in computer vision proves the importance of modeling global dependencies. Inspired by [27,41], we present a Transformer interaction fusion module (TIFM) to explore the rich contextual information between features. The input of TIFM includes the modality-specific features and the aggregated feature (the output of the FSM). It should be noted that the aggregated feature and modality-specific features contain similar information (such as background information), and reducing redundant information can improve fusion efficiency. To solve this problem, a 1 × 1 convolution layer and sigmoid activation function are used to form the gated network to adjust the input adaptively.

As shown in Figure 4, to reduce the complexity of the model, we remove the position encoding and feed-forward network of the original Transformer. Each input feature is linearly transformed to produce a query matrix *Q*, a key matrix *K*, and a value matrix *V*. Through *Q*, *K* generates attention weights to modulate *V*. *Q* is then added to the output of Attention as a residual. Attention part can be expressed as follows:(4)$$Attention\left(Q,K,V\right)=softmax\left({\frac{QK}{\sqrt{C}}}^{T}\right)V$$
where *C* is the dimension of key matrix to normalize attention. As we all know, the product of *Q* and *K* transpose can be regarded as the inner product of the corresponding vector so that the dependencies between any two elements of the global context can be modeled. Therefore, the encoders highlight the critical information by modeling its element dependencies. The decoders learn long-range dependencies among features to further enhance semantic information.

As shown in Figure 5, TIFM includes two encoders and two decoders. The encoders perform self-enhancement, and the decoders realize the interactive enhancement of the aggregated features (Xsel) and modality-specific features (Xvis and Xinf). Then, the element-wise summation operation combines the multiple enhancement features into a unified feature, and an encoder is connected to further enhance the semantic information. Specifically, we use the aggregated feature as the query for all features, resulting in a self-enhancement module (*SE*) and two feature interactive enhancement modules (*IE*). The whole process can be expressed as follows:(5)XΘ=SESEXsel,Xsel+IEXsel,Xvis+IEXsel,Xinf
where XΘ represents the interactive enhanced features. Where Xsel,Xvis,Xinf∈ℝC×H×W are the three inputs of TIFM. *C*, *H*, and *W* represent the number of channels, height, and width of the feature matrix.

### 3.4. Global Feature Fusion Module

In RGBT tracking, many algorithms aggregate the output features in the way of concatenating, which ignores the different contributions of different modalities in various scenarios. To weigh the contribution of the two modalities, we designed the global feature fusion module (GFM). The GFM aggregates multi-modalities features at the channel and spatial levels. As shown in Figure 6, it includes two branches, namely the channel integration branch and the spatial integration branch.

A channel integration branch similar to FSM is designed to obtain the channel weights of the two modalities. The difference from FSM is that the two input features are concatenated here, and GAP and global maximum pooling (GMP) are used to obtain global information and salient information. Finally, two fully connected layers and a softmax function to transform the channel vector nonlinearly obtain the channel weight $w_{c}$.
(6)$$w_{c}=\phi_{2}(\phi_{1}\left(GAP\left(P^{a}\right)+GMP\left(P^{a}\right)\right)$$
(7)$$P^{a}=cat\left(x_{i},x_{v}\right)$$
where $\phi_{1}$, $\phi_{2}$ denote the fully connected layer. $P^{a}$ represents the concatenated feature. $x_{i}$ and $x_{v}$ represent the output of the third layer.

At the same time, we design a spatial integration branch, which uses the max function to obtain the maximum value of the corresponding channel dimension of each spatial position and uses a 3 × 3 convolution smoothing feature to get the spatial weight $w_{s}$. $w_{s}$ evaluates the contribution of each spatial location while highlighting candidates and suppressing distractions.
(8)$$w_{s}=conv_{3\times 3}\left(max(P^{a}\right))$$

After the channel and spatial weights are obtained, they are fixed between 0 and 1 through softmax function σ. Additionally, we multiply them to get a 3D feature matrix with the same size as the original feature, then multiply the 3D feature with $P^{a}$. Finally, the element-wise summation of the $P^{a}$ is carried out to get the optimized feature P. The mathematical expression is given as follows:(9)$$P=P^{a}⊗\left[\sigma \left(w_{s}\right)⊗\sigma \left(w_{c}\right)\right]+P^{a}$$

## 4. Experiments

To verify the effectiveness of our proposed method, we conducted many representative experiments. The experimental environment is configured as follows: NVIDIA GeForce RTX3090 GPU server, PyTorch 1.12, and Python 3.8.

### 4.1. Implementation Details

#### 4.1.1. Network Training

In this work, we train MSIFNet in multiple steps, and stochastic gradient descent (SGD) algorithm is adopted to optimize MSIFNet. During training, we use softmax cross-entropy loss function for binary classification. The weight decay and momentum are set to 0.0005 and 0.9, respectively. The first step is to train the backbone network. The backbone network is initialized by the pre-trained VGG-M model on ImageNet [42], and the specific-modality branch and the full connection layer are trained with 100 epoch iterations. The learning rate of the convolution layer and the full connection layer is 0.0005 and 0.001, respectively. In the second step, we loaded the training parameters of the first step, randomly initialized the parameters of the multi-scale branches and FSM, and conducted the training of 500 epoch iterations. The learning rate was consistent with that of the first step. In the third stage, we load the training parameters of the second step and randomly initialize TIFM and GFM to train 1000 epoch iterations, and the learning rate of TIFM is 0.001, and the others are 0.0005. In each iteration, eight frames of images are randomly obtained from the video sequence. Gaussian samples are performed on these 8 frames to get 256 positive and 768 negative samples, of which 32 positive samples and 96 negative samples are generated for each frame image. It is considered a positive sample when the ratio of overlap rate (IoU) between the sample and the ground truth is [0.7, 1], while a sample in the range of [0, 0.5] is considered a negative sample. Notably, we train MSIFNet with the GTOT [43] dataset and test it on RGBT234 [44] and LasHeR [45] datasets. When testing on GTOT, we randomly selected 50 video sequences on RGBT234 for training.

#### 4.1.2. Online Tracking

Like MDNet [46], we freeze all network parameters except Fc4-Fc6 and randomly initialize a new FC6 branch during tracking. Then, Gaussian sampling is performed on the target bounding box in the first frame, and 500 positive samples and 5000 negative samples are obtained as training sets for 50 epoch iterations of fine-tuning FC4, FC5, and FC6. The learning rate for Fc4 and Fc5 is 0.0001, and for Fc6 is 0.001. To make the tracking results more accurate, we update the parameters of the three fully connected layers in the short- and long-term and use the bounding box regression technique to solve the target scale change problem. Given the unreliability of subsequent frames, we only collected 1000 samples in the first frame to train the regression factor. For the t frame, the network uses the tracking result of the t-1 frame to sample 256 candidate samples and send them to the network to predict the target state. We select the five candidate samples with the highest score and take their average as the tracking result of the current frame. The tracking is successful when the result score exceeds 0, and the results are adjusted using a bounding box regressor for more accurate positioning. The network repeats until the entire sequence is tracked.

### 4.2. Result Comparisons

#### 4.2.1. Datasets and Evaluation Metrics

(1)GTOT dataset: the GTOT dataset contains 50 pairs of RGBT video sequences collected in different scenarios and conditions, aligned spatially and temporally, totaling about 15K frames. According to the target state, it is divided into 7 challenge attributes to analyze the performance of the tracker under different conditions.(2)RGBT234 dataset: it is a large dataset after the RGBT210 [47] dataset, adding 34 video sequences based on RGBT210. It includes 234 pairs of highly aligned RGBT video sequences and 12 challenge attributes for approximately 234K frames. It provides more accurate annotations and considers the challenges posed by various environments.(3)LasHeR dataset: the LasHeR dataset is a more comprehensive and extensive RGBT dataset containing 1224 pairs of aligned video sequences and 19 attribute annotations. Among them, 245 sequences were selected as the test dataset, and the rest were used for training.

In the above datasets, precision rate (PR) and success rate (SR) are commonly used to evaluate tracker performance. Specifically, PR represents the percentage of frames for which the Euclidean distance between the tracking result and the ground truth is below the set threshold. Since the target of the GTOT is smaller, the threshold is set to 5 pixels. We set the threshold to 20 pixels for RGBT234 and LasHeR datasets. SR measures the percentage of frames where the overlap ratio between the tracking result and the ground truth is greater than the set threshold. Normalized PR (NPR) [48] is also commonly used to evaluate the LasHeR dataset.

#### 4.2.2. Evaluation of GTOT Dataset

Overall Comparison: To verify the effectiveness of MSIFNet on the GTOT dataset, we compared the tracking results with some advanced RGBT tracking algorithms, including M^5^L [49], CAT [12], APFNet [38], MANet [11], HDINet [50], DAFNet [35], MACNet [51], SGT [47], ECO [52], MDNet+RGBT, RT-MDNet [53], etc. The results of the PR and SR evaluations are shown in Figure 7. Specifically, MSIFNet gains 10.4% in PR and SR compared to the baseline tracker MDNet + RGBT. Compared to the recent APFNet, MSIFNet is 0.4% higher on SR and only 0.1% lower on PR. Compared to HDINet and CAT trackers, our trackers have a performance improvement of 1.6%/2.3% and 1.5%/2.4% on PR/SR, respectively. In summary, compared with some advanced RGBT algorithms, the effectiveness of our algorithm is proven.

Attribute-based performance: To evaluate the specific performance of our algorithm in addressing challenges, we compared the results of MSIFNet with those of other state-of-the-art RGBT trackers on various challenge attributes. The GTOT dataset contains 7 challenge attribute labels, including occlusion (OCC), large scale variation (LSV), a small object (SO), fast motion (FM), deformation (DEF), low illumination (LI), and thermal crossover (TC). The evaluation results are presented in Table 1. The best result is indicated in red, and the second and third are green and blue, respectively. The comparative results show that the overall performance of MSIFNet is optimal, especially in solving challenges such as OCC, LSV, FM, TC, and SO, which fully demonstrates the effectiveness of our method. In particular, MSIFNet ranks first for performance in handling large-scale changes and small object challenges, which proves the effectiveness of our proposed design of multi-scale information mining. MSIFNet is first place in both PR and SR on OCC, demonstrating that the GFM has a solid ability to highlight important information about the target and suppress the background.

#### 4.2.3. Evaluation on RGBT234 Dataset

Overall Comparison: As shown in Figure 8, a comprehensive evaluation was performed on the RGBT234 dataset. The algorithm in this paper also achieves advanced performance. Compared with JMMAC [54], the proposed algorithm improves PR by 2.7%, while SR is only 0.3% lower. In addition, compared to CAT, M^5^L and HDINet, MSIFNet has a performance improvement of 1.3%/0.9%, 2.2%/2.8%, and 3.4%/1.1% on PR/SR, respectively. However, compared to high-performance CMPP [36], MSIFNet has a certain gap. The reason is that the proposed algorithm only uses the information of the current frame to model the target appearance, while CMPP enhances the representation of the current frame with the help of a large amount of historical information.

Attribute-based performance: At the same time, we also compared MSIFNet with 5 excellent trackers on 12 challenge attributes on the RGBT234 dataset. Challenging attributes include scale variation (SV), partial occlusion (PO), deformation (DEF), background clutter (BC), low resolution (LR), low illumination (LI), camera movement (CM), fast motion (FM), heavy occlusion (HO), no occlusion (NO), motion blur (MB), and thermal crossover (TC). As can be seen from Table 2, MSIFNet performs well in the challenges of HO, SV, LI, LR, TC, MB, DEF, CM, and FM. Notably, MSIFNet ranked in the top three across all challenges. The PR/SR metric on SV is 82.0%/57.6% compared to state-of-the-art RGBT trackers, proving that MSIFNet can adapt well to scale changes and mine multi-scale clues well. Moreover, the performance of our algorithm on LI is also excellent, which proves that the FSM and TIFM can achieve modality information interaction and fusion well. The FSM and TIFM make the RGB modality have the infrared modality information, which can cope with the challenge of low illumination well.

#### 4.2.4. Evaluation of LasHeR Dataset

To further demonstrate the robustness of MSIFNet to different datasets, we evaluate the performance of the tracker on LasHeR. The comparison results of NPR and SR indicators are shown in Figure 9. Compared to 11 RGBT algorithms, MSIFNet obtained the best results in NPR and SR. Specifically, MSIFNet outperformed the best-performing MaCNet on NPR/SR, up 0.8% and 0.2%, respectively. In addition, MSIFNet has a 3.3%/3.8% and 2.4%/3.8% improvement in NPR and SR compared to CAT and MANet++ [55]. These results demonstrate the robustness of MSIFNet to different datasets.

### 4.3. Qualitative Analysis

In Figure 10, we perform a qualitative analysis of our algorithm. We compared MSIFNet with four advanced RGBT algorithms on six pairs of video sequences, namely children4, mancross1, diamond, Yellowcar, single3, and Night2. In the case of the first children4 sequence target moving the fast and low resolution, the target bounding box of all other trackers gradually drifts, and the accuracy of MSIFNet is barely affected. In the mancross1 sequence and diamond sequence, there are challenging attributes such as background clutter, scale variation, and heavy occlusion. Other algorithms keep the target in the lost state after heavy occlusion, but our method can track the object in a good state after occlusion. Meanwhile, in the Yellowcar sequence, our tracker can still maintain almost the same size as the ground truth box as the target scale changes. Our tracker can also robustly track the target for the single3 and Night2 sequences with a small object, fast motion, low illumination, background clutter, and partial occlusion attributes. It is clear that MSIFNet can locate the target accurately during the tracking process and is capable enough to cope with various challenges.

### 4.4. Ablation Study

To verify the validity of the proposed modules, ablation experiments were performed on the RGBT234 dataset. We compared the tracking performance of the following methods: (1) MSIFNet-FSM, which removes the FSM and performs simple addition operations for multi-scale features; (2) MSIFNet-TIFM, which removes the TIFM and adds the output of FSM directly to the backbone features; (3) MSIFNet-GFM, which removes the GFM and directly concatenates the output features of the third layer; and (4) MSIFNet-Gate, which removes the Gated network and inputs the TIFM for interactive fusion without gating. As seen from Table 3, MSIFNet-FSM has the lowest performance, mainly because a large amount of redundant information is inevitably introduced when aggregating data from multiple branches, which proves the necessity of introducing aggregation modules into multi-scale branches. The performance of MSIFNet is significantly better than that of MSIFNet-TIFM, which verifies that the proposed TIFM can effectively enhance semantic information by learning long-distance dependencies. The accuracy of MSIFNet-GFM and MSIFNet-Gate is somewhat reduced in MSIFNet, which also demonstrates the effectiveness of the GFM and Gate network.

We compared the performance of the following variants to verify the effectiveness of the multi-scale features extracted from each layer. (1) MSIFNet-L1 means deleting the 5 × 5 and 3 × 3 branches of the first layer and directly fusing the backbone features while leaving the other layers unchanged. (2) MSIFNet-L2 means that the 3 × 3 and 1 × 1 branches of the second layer are deleted, while others remain unchanged. (3) MSIFNet-L3 means to delete the convolution branch of 1 × 1 in the third layer. The results of the comparison are shown in Table 4. Experimental results show that the performance of MSIFNet is better than that of MSIFNet-L1, MSIFNet-L2, and MSIFNet-L3, which proves the effectiveness of multi-scale features in each layer.

## 5. Conclusions

In this paper, we propose a multi-scale feature interaction fusion network (MSIFNet), which can mine the multi-scale information of RGB and infrared images to better identify targets of different sizes and further improve tracking performance. In particular, we use a feature selection module to adaptively select features from multiple branches, and the Transformer interactive fusion module is used to mine complementary information between features and enhance semantic representation. Moreover, a global feature fusion module is designed to adjust the global information in spatial and channel dimensions, respectively. Numerous experiments on publicly available GTOT, RGBT234, and LasHeR datasets have shown that our MSIFNet has advanced performance in handling various challenges. In the future, we will further explore multi-modalities fusion mechanisms to achieve more robust feature representation.

## Figures and Tables

**Figure 1 sensors-23-03410-f001:**
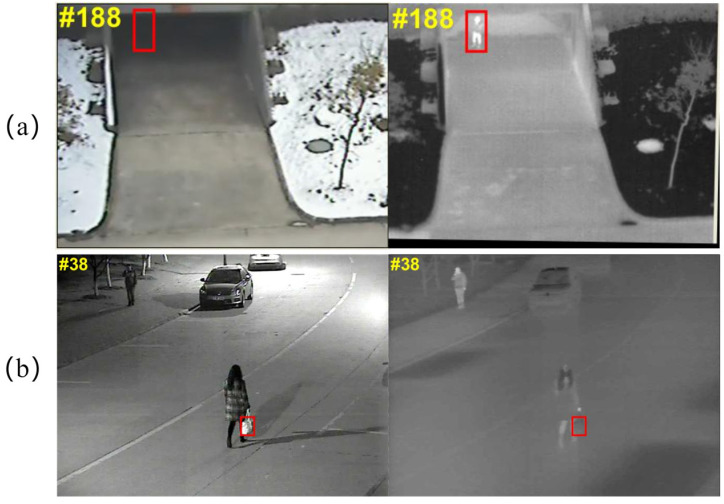
Illustration of the advantages of RGB and infrared modalities. (**a**) The advantage of thermal infrared modality over RGB modality. (**b**) The advantage of RGB modality over thermal infrared modality.

**Figure 2 sensors-23-03410-f002:**
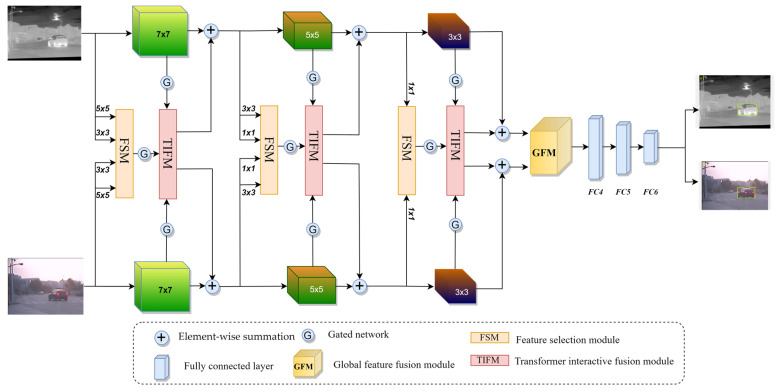
The overview of the proposed MSIFNet framework.

**Figure 3 sensors-23-03410-f003:**
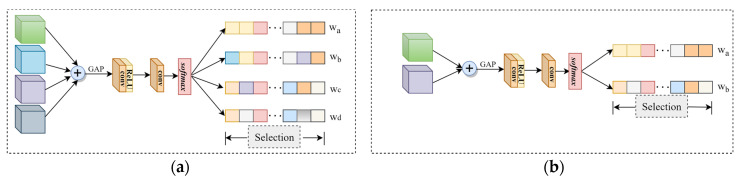
Specific structure of the FSM. The structure of the FSM of the first and second layers is shown in (**a**), and the third layer is shown in (**b**).

**Figure 4 sensors-23-03410-f004:**
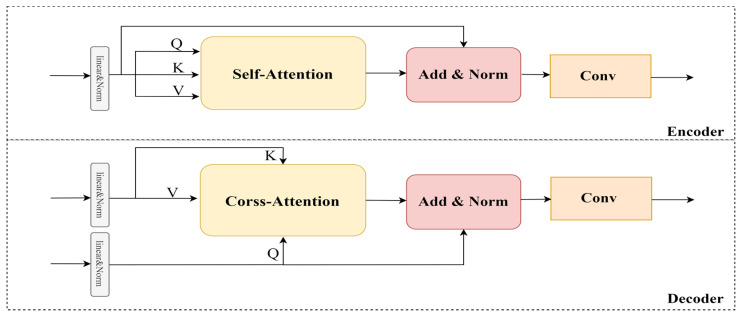
Specific structure of the encoder and decoder.

**Figure 5 sensors-23-03410-f005:**
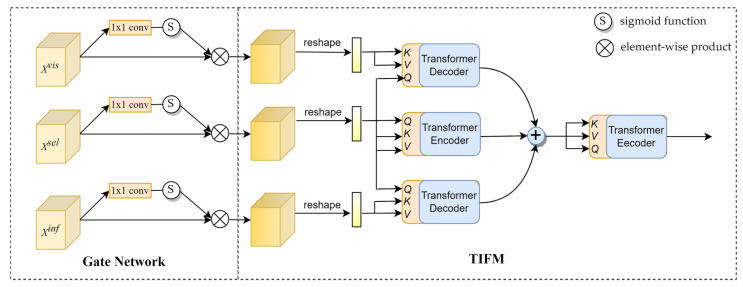
Specific structure of the Gate Network and TIFM.

**Figure 6 sensors-23-03410-f006:**
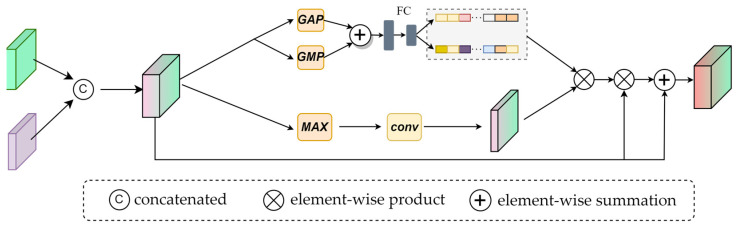
Specific structure of the GFM.

**Figure 7 sensors-23-03410-f007:**
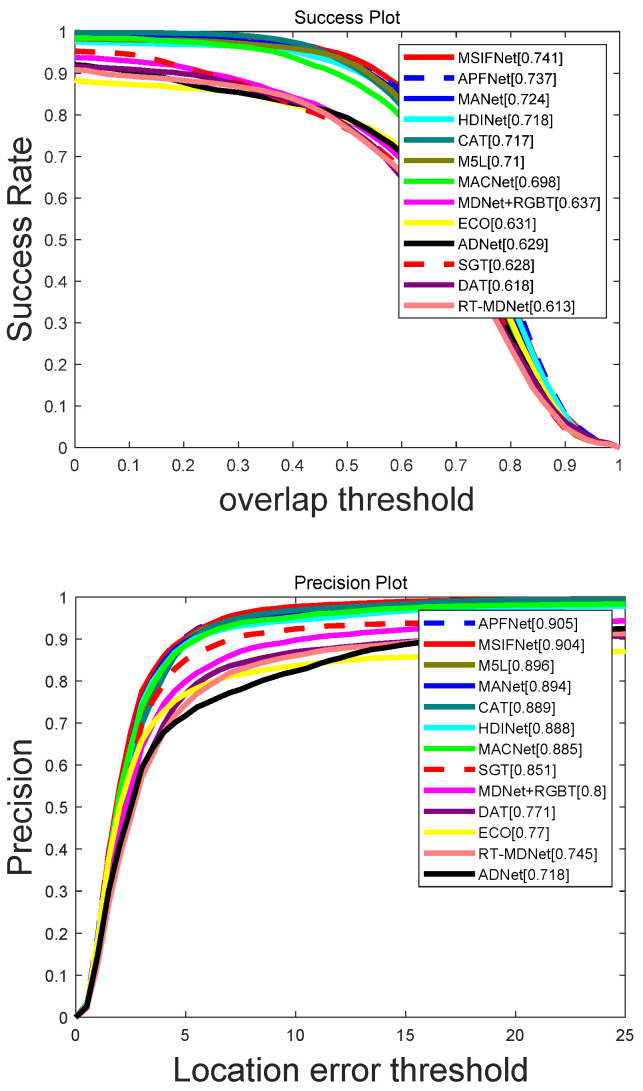
Evaluation curves for PR/SR metrics on GTOT.

**Figure 8 sensors-23-03410-f008:**
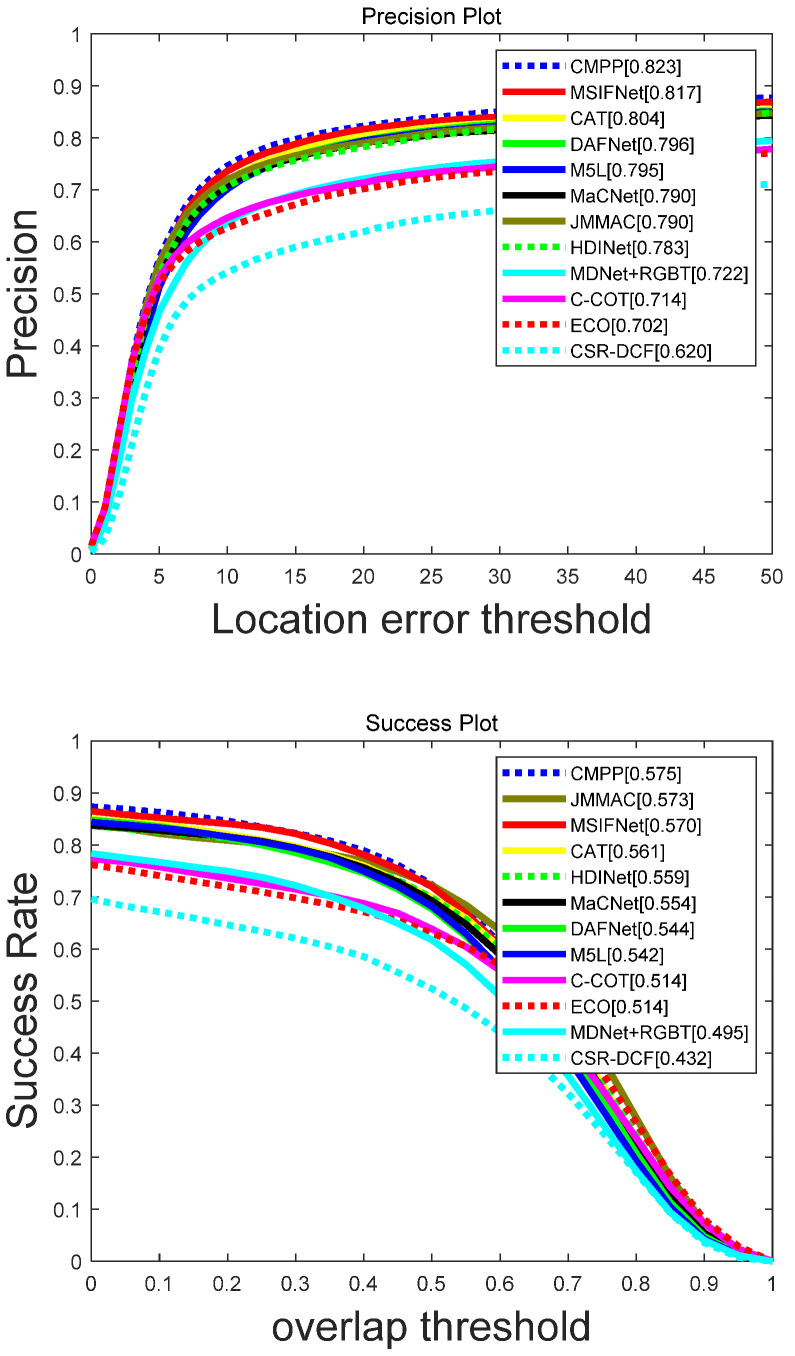
Evaluation curves for PR/SR metrics on RGBT234.

**Figure 9 sensors-23-03410-f009:**
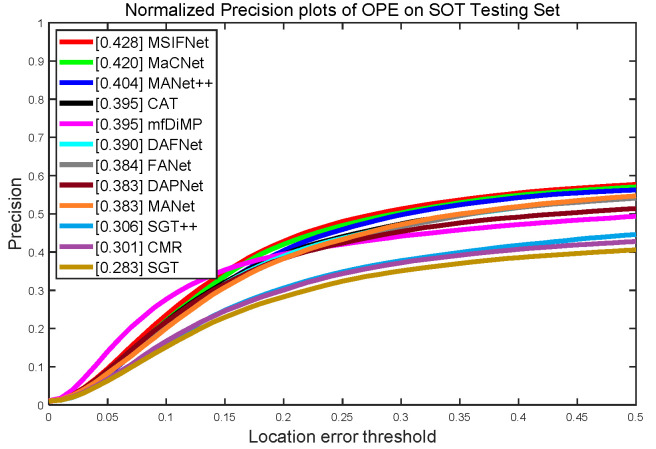
Evaluation curves for NPR/SR metrics on LasHeR.

**Figure 10 sensors-23-03410-f010:**
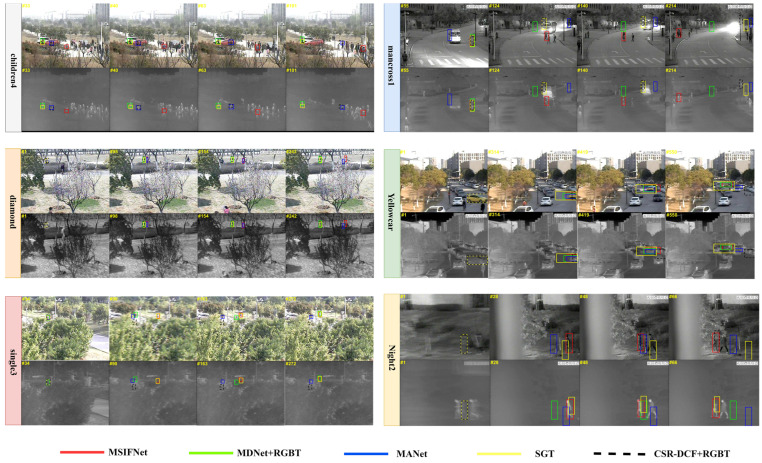
Qualitative comparison with four excellent RGBT trackers.

**Table 1 sensors-23-03410-t001:** Attribute-based PR/SR scores (%) on the GTOT dataset.

	SGT	MDNet + RGBT	MANet	CAT	APFNet	MSIFNet
OCC	81.0/56.7	82.9/64.1	88.2/69.6	89.9/69.2	90.3/71.3	90.9/72.2
LSV	84.2/54.7	77.0/57.3	86.9/70.6	85.0/67.9	87.7/71.2	88.0/72.0
FM	79.9/55.9	80.5/59.8	87.9/69.4	83.9/65.4	86.5/68.4	88.2/70.2
LI	88.4/65.1	79.5/64.3	91.4/73.6	89.2/72.3	91.4/74.8	91.2/74.5
TC	84.8/61.5	79.5/60.9	88.9/70.2	89.9/71.0	90.4/71.6	91.4/72.9
SO	91.7/61.8	87.0/66.2	93.2/70.0	94.7/69.9	94.3/71.3	95.9/72.3
DEF	91.9/73.3	81.6/68.8	92.3/75.2	92.5/75.5	94.6/78.0	92.8/77.5
ALL	85.1/62.8	80.0/63.7	89.4/72.4	88.9/71.7	90.5/73.7	90.4/74.1

**Table 2 sensors-23-03410-t002:** Attribute-based PR/SR scores (%) on the RGBT234 dataset.

	MDNet + RGBT	DAFNet	M^5^L	CAT	HDINet	MSIFNet
NO	86.2/61.1	90.0/63.6	93.1/64.6	93.2/66.8	88.4/65.1	92.6/66.0
PO	76.1/51.8	85.9/58.8	86.3/58.9	85.1/59.3	84.9/60.4	84.7/59.5
HO	61.9/42.1	68.6/45.9	66.5/45.0	70.0/48.0	67.1/47.3	73.8/50.5
LI	67.0/45.5	81.2/54.2	82.1/54.7	81.0/54.7	77.7/53.2	83.2/55.7
LR	75.9/51.5	81.8/53.8	82.3/53.5	82.0/53.9	80.1/54.5	85.1/56.5
TC	75.6/51.7	81.1/58.3	82.1/56.4	80.3/57.7	77.2/57.5	84.4/59.0
DEF	66.8/47.3	74.1/51.6	73.6/51.1	76.2/54.1	76.2/56.5	74.7/53.5
FM	58.6/36.3	74.0/46.5	72.8/46.5	73.1/47.0	71.7/47.5	74.7/47.5
SV	73.5/50.5	79.1/54.4	79.6/54.2	79.7/56.6	77.5/55.8	82.0/57.6
MB	65.4/46.3	70.8/50.0	73.8/52.8	68.3/49.0	70.8/52.6	75.8/54.4
CM	64.0/45.4	72.3/50.6	75.2/52.9	75.2/52.7	69.7/51.4	77.3/54.9
BC	64.4/43.2	79.1/49.3	75.0/47.7	81.1/51.9	71.1/47.8	80.5/52.3
ALL	72.2/49.5	79.6/54.4	79.5/54.2	80.4/56.1	78.3/55.9	81.7/57.0

**Table 3 sensors-23-03410-t003:** The PR/SR scores of our method were compared with variants on the RGBT234 datasets to verify the validity of the proposed modules.

		MSIFNet-FSM	MSIFNet-TIFM	MSIFNet-GFM	MSIFNet-Gate	MSIFNet
RGBT234	PR	0.799	0.806	0.813	0.811	**0.817**
	SR	0.557	0.561	0.564	0.565	**0.570**

**Table 4 sensors-23-03410-t004:** The PR/SR scores of our method were compared with variants on the RGBT234 datasets to verify the validity of the multi-scale features extracted from each layer.

		MSIFNet-L1	MSIFNet-L2	MSIFNet-L3	MSIFNet
RGBT234	PR	0.801	0.806	0.814	**0.817**
	SR	0.563	0.563	0.568	**0.570**

## Data Availability

Public datasets.

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
