# Peer review of "Multi-Scale Feature Interactive Fusion Network for RGBT Tracking"

_sensors, 2023, doi:10.3390/s23073410_

Round 1

Reviewer 1 Report

This paper proposed a multi-scale feature interactive fusion network (MSIFNet) to use the multi-scale information of RGB images and thermal infrared images to further improve the performance of target tracking. The proposed method has a selection module to adaptively select features from multiple branches, and use Transformer to achieve feature enhancement. Experiments show its superior performance. 

This paper shows efforts from authors to tie all the things together and make them work. However, the improvement introduced by the proposed method is not too much. The authors of this paper are suggested to explain the contributions of the proposed method to this area of tracking using images. What is special? Instead of showing it uses Transformer and the common equations of Q, K, and V, the authors are encouraged to explain the internal principle and significance of the proposed approach. In addition, showing what would happen if no such mechanisms.    

The presentation of the paper needs to be improved. There are many typos, grammar errors, and unclear sentences. BTW, the authors should explain some acronyms before using them, e.g., RGBT, GFM, etc. 

Author Response

We appreciate the time and effort that you dedicated to providing feedback on our manuscript and are grateful for the insightful comments and valuable improvements to our paper. Below, I will reply to your questions one by one.

Point 1: However, the improvement introduced by the proposed method is not too much. The authors of this paper are suggested to explain the contributions of the proposed method to this area of tracking using images. What is special? 

Response 1: Thank you for the above valuable suggestions. The introduction of our manuscript did not clearly describe the significance and innovation of the research. In light of this, we have strengthened the Introduction and Related work sections to bring out the contributions and innovations of this paper. The contributions and innovations of this paper are:

(1) In the field of image tracking, we propose a new multi-scale feature interactive fusion network (MSIFNet) to implement robust RGBT tracking. The network can improve the recognition ability of targets of different sizes by fully exploiting multi-scale information, thus improving tracking accuracy and robustness. 

(2) We design a feature selection module that adaptively selects multi-scale features for fusion by the channel-aware mechanism while effectively suppressing noise and redundant information brought by multiple branches.

(3) We propose a Transformer interactive fusion module to further enhance the aggregated feature and modality-specific features. It improves long-distance feature association and enhances semantic representation by exploring rich contextual information between features.

(4) We design a global feature fusion module, which adaptively adjusts the global information in spatial and channel dimensions, respectively, so as to integrate the global information more effectively.

Point 2: Instead of showing it uses Transformer and the common equations of Q, K, and V, the authors are encouraged to explain the internal principle and significance of the proposed approach.

Response 2: Sorry for neglecting this question while writing. We have supplemented the internal principle in the second paragraph of Section 3.3 of the article, as follows:

As we all know, the product of Q and K transpose can be regarded as the inner product of the corresponding vector so that the dependencies between any two elements of the global context can be modeled. Therefore, the encoders highlight the key position information by modeling its element dependencies. The decoders learn long-range dependencies among features to further enhance semantic information.

Point 3: In addition, showing what would happen if no such mechanisms.

Response 3: Thank you for pointing out this problem in the manuscript. For this problem, we have summarized each module as follows:

 If there is no FSM, it will lead to information redundancy, because a large amount of redundant information is inevitably introduced when aggregating data from multiple branches, and FSM can adaptively activate valuable features for fusion while effectively suppressing noise and redundant information. Without TIFM, some semantic information is lost, because TIFM can further enhance semantic representation by establishing long-distance dependencies. For the GFM, it can weigh the contribution of different modalities in different scenarios. In addition, we conducted corresponding ablation experiments in Section 4.4, which showed the effectiveness of each module.

Point 4: The presentation of the paper needs to be improved. There are many typos, grammar errors, and unclear sentences. BTW, the authors should explain some acronyms before using them, e.g., RGBT, GFM, etc.

Response 4: Sorry for some problems with my English expression earlier. We have carefully revised this manuscript to improve the readability of the paper. For acronyms, we have added explanations to acronyms and carefully checked all acronyms.

Reviewer 2 Report

The authors proposed a fusion network with an interactive feature embedded for RGBT tracking. The results look promising and interesting. I would like to recommend it be published on Sensors if the authors can address the following questions. 

1. The abstract should be concise. 

2. The abbreviation RGBT is not explained when first appears in the context. 

3. Relevant references should be cited when introducing pixel lever, feature level, and decision level fusion. 

4. What was the platform used and how long it took to train the network compared to other networks?

Author Response

We appreciate the time and effort that you dedicated to providing feedback on our manuscript and are grateful for the insightful comments and valuable improvements to our paper. Below, I will reply to your questions one by one.

Point 1: The abstract should be concise.

Response 1: Thank you for your reminder. We have revised the Abstract and modified the redundant sentences to make them more concise.

Point 2: The abbreviation RGBT is not explained when first appears in the context.

Response 2: Thank you so much for your careful check. We have added explanations to this abbreviation and carefully checked all abbreviations.

Point 3: Relevant references should be cited when introducing pixel lever, feature level, and decision level fusion.

Response 3: Thank you for pointing out this problem in the manuscript. We added corresponding references when introducing pixel lever, feature level and decision level fusion, and carefully detected and revised oversights elsewhere.

Point 4: What was the platform used and how long it took to train the network compared to other networks?

Response 4: Sorry for neglecting this question while writing. We supplement the platform used and the time taken to train the network in Section 4. Specifically, the experimental environment is configured as follows: NVIDIA GeForce RTX3090 GPU server, PyTorch 1.12 and Python 3.8. During the three-stage training, 100 epoch iterations are trained in the first stage, 500 epoch iterations are trained in the second stage, and 1000 epoch iterations are trained in the third stage.

Round 2

Reviewer 1 Report

The authors have addressed most of the previous comments.